# Dynamic Genome Editing Using In Vivo Synthesized Donor ssDNA in *Escherichia coli*

**DOI:** 10.3390/cells9020467

**Published:** 2020-02-18

**Authors:** Min Hao, Zhaoguan Wang, Hongyan Qiao, Peng Yin, Jianjun Qiao, Hao Qi

**Affiliations:** 1School of Chemical Engineering and Technology, Tianjin University, Tianjin 300072, China; min1213@tju.edu.cn (M.H.); wzg1895@tju.edu.cn (Z.W.); qhy_@tju.edu.cn (H.Q.); yp_2019207614@tju.edu.cn (P.Y.); jianjunq@tju.edu.cn (J.Q.); 2Key Laboratory of Systems Bioengineering of Ministry of Education, Tianjin University, Tianjin 300072, China; 3SynBio Research Platform, Collaborative Innovation Center of Chemical Science and Engineering, Tianjin University, Tianjin 300072, China

**Keywords:** rolling circle replication, rolling circle origin, single-strand DNA, guide RNA, PAM-independent, SpyCas9

## Abstract

As a key element of genome editing, donor DNA introduces the desired exogenous sequence while working with other crucial machinery such as CRISPR-Cas or recombinases. However, current methods for the delivery of donor DNA into cells are both inefficient and complicated. Here, we developed a new methodology that utilizes rolling circle replication and Cas9 mediated (RC-Cas-mediated) in vivo single strand DNA (ssDNA) synthesis. A single-gene rolling circle DNA replication system from Gram-negative bacteria was engineered to produce circular ssDNA from a Gram-positive parent plasmid at a designed sequence in *Escherichia coli*. Furthermore, it was demonstrated that the desired linear ssDNA fragment could be cut out using CRISPR-associated protein 9 (CRISPR-Cas9) nuclease and combined with lambda Red recombinase as donor for precise genome engineering. Various donor ssDNA fragments from hundreds to thousands of nucleotides in length were synthesized in *E. coli* cells, allowing successive genome editing in growing cells. We hope that this RC-Cas-mediated in vivo ssDNA on-site synthesis system will be widely adopted as a useful new tool for dynamic genome editing.

## 1. Introduction

After several decades of development, genome engineering has become a highly developed and indispensable tool for biological study and bioengineering applications [1]. A large toolbox has been developed using different molecular machinery, including recombinases, zinc finger nucleases (ZFNs), transcription activator-like effector nucleases (TALENs) and the clustered regularly interspaced short palindromic repeats (CRISPR) and CRISPR-associated protein 9 (Cas9) systems (CRISPR/Cas9 system) [2,3,4]. Homologous recombination can introduce precise deletions or insertions into the genome. The donor DNA template, either single-stranded DNA (ssDNA) or double-stranded DNA (dsDNA), is a crucial part for homology-directed genome editing [5,6]. Recombineering mediated by linear dsDNA fragments or circular plasmids requires long (1–5 kb) homology arms flanking the insertion sequence [7,8]. By contrast, commercial ssDNA oligos can only provide short homology arms (<200 nts) [9,10]. Experiments with single-stranded donor DNA yielded between 10 and 100% recombination frequency for point mutations [7]. Oligo-mediated recombination can be used to induce allele substitution, small (20–30 base) insertions [11,12] or deletions of up to ~45 kb [13]. Although short ssDNA oligo are useful for the modification of multiple loci, they cannot be readily used for large knock-in applications [14]. It is known that longer sequences better accommodate the extended spacing at editing sites [15]. Long linear ssDNAs synthesized in vitro have been used for homology-directed genome editing [16,17]. For genomic knock-ins in human cell lines, long (~2 kb) ssDNA templates have unique advantages in terms of repair specificity, while dsDNA templates of a similar size (polymerase chain reaction (PCR) products or plasmids) suffer from high rates of off-target integration [18]. Furthermore, the editing efficiency for the integration of larger sequences into the genome is heavily dependent on the concentration of introduced donor DNA and the transformation efficiency, which often require laborious and outdated genetic engineering techniques [7,19]. Although there are still many unknown elements in the mechanism of homologous recombination, it is well established that ssDNA is often more favorable than dsDNA as donor template with regard to biological function. [15,18]. However, the synthesis of ssDNA fragments several kilobases in length requires very expensive and complicated synthesis processes. [20,21,22]

Theoretically, in-vivo synthesis of an ssDNA donor template in the cell could be of great advantage for genome editing, especially for knock-ins of exogenous sequences. Fortunately, in-vivo ssDNA synthesis is known to occur in nature under specific circumstances. Notably, bacteriophages such as M13, fd and f1 have ssDNA genomes, and phagemids based on their backbones can replicate as plasmids inside the cell [23,24,25]. In fact, phage systems were some of the first tools for the production of ssDNA. Alternatively, rolling-cycle replication (RCR) of plasmids has been shown to produce intermediates in the form of circular ssDNA. A number of small, multi-copy rolling-circle plasmids from *Staphylococcus aureus* and *Bacillus subtilis* have been studied in detail [26,27,28,29,30,31]. The replication of these RCR plasmids requires only three elements—a gene encoding the initiator protein (Rep), a double strand origin (dso), and a single strand origin (sso) [32], greatly simplifying their genetic engineering. In particular, the RCR plasmid pC194 has been demonstrated to replicate effectively in *Escherichia coli* and generate a circular ssDNA in this host [33,34]. Furthermore, there is evidence that a sequence of only 55 bp is required to initiate leading-chain replication, which can be terminated perfectly at the 36 bp sequence position [35]. It has been shown that ssDNA can be produced using a reverse transcriptase and directly applied for genome editing in vivo [36]. However, only short ssDNAs were synthesized using this system, while the production and application of longer ssDNAs was not reported.

Here, a rolling-circle replication system driven by a single gene from Gram-positive bacteria was engineered to generated large circular ssDNAs from its parent Gram-negative plasmid vector in a highly controlled fashion in *E. coli*. A specific sequence cassette was designed to tightly control the stable generation of circular ssDNA, from which a desired fragment was cut out using the SpyCas9 nuclease in a protospacer adjacent motif (PAM)-independent manner. Therefore, only circular ssDNA could be cleaved but not the dsDNA plasmid vector, while both of them carry the same target sequence. In *E. coli*, it was demonstrated that circular ssDNAs ranging from 581 to 4960 nts can be produced in vivo, and linearized ssDNA fragments were able to function as donor templates for lambda-Red mediated homologous recombination at target loci on the *E. coli* genome. Using a linear ssDNA fragment with a length of 1208 nts, a 1011 bp sequence was successfully inserted in the middle of the *lacZ* gene in place of an original 11bp sequence. To the best of our knowledge, this is the largest ssDNA fragment generated inside a cell for genome editing. Moreover, this rolling circle replication and Cas9 mediated(RC-Cas-mediated) ssDNA synthesis system can remain functional in the cell for long periods and even be passed on to the next generation. Accordingly, serial subculture significantly improved the genome editing efficiency. These novel features offer distinct advantages compared to transformation with DNA fragments from the outside of a cell, and more biological functions can potentially be coupled together with desired timing. Because the system comprises only two enzymes, we believe that it can be applied to other organisms. Therefore, this in vivo RC-Cas linear ssDNA synthesis system offers a new method for dynamic multiplex genome editing.

## 2. Materials and Methods

### 2.1. Strains and Culture Conditions

Wild-type *E. coli* strain DH5α, BL21, DH10β and MG1655 were used in this study. Those cells were cultured in lysogeny broth (LB; 1% (*w*/*v*) tryptone, 0.5% (*w*/*v*) yeast extract, 1% (*w*/*v*) NaCl) with shaking at 220 rpm or on LB agar plates. For ssDNA expression, overnight culture of *E. coli* cells harboring corresponding plasmid were inoculated to fresh LB medium with appropriate antibiotic in 1:100 dilution, then it was cultured at 37 °C with shaking 220 rpm. To induce RepH protein expression, L^+^Rha was added to the culture when the OD_600_ reached 0.5. Cells were harvested at mid-log phase (OD_600_~1.0), then purified the plasmids by plasmid Mini-preparation Kit (DP103, TIANGEN (Beijing) or lysed by cell lysis solution. For genome editing, *E. coli* MG1655 competent cells harboring pRedCas9 which was induced by isopropyl-beta-D-thiogalactopyranoside (IPTG) for bacteriophage λ Red protein (λ-RED) proteins expressed. The cells with or without plasmids were grown in LB medium supplemented with the appropriate antibiotics at 30 °C. Bio-Rad Micro Pulser (0.1 cm cuvette, 1.80 kV) was used for electroporation. 1 µg of plasmid DNA or 1 µM of the editing cassette was mixed with 50 µL competent cells for electroporation. Cells after electroporation were immediately added into 2 mL LB and recovered for 2 h at 30 °C, and inoculated to fresh LB medium in 1:100 dilution or spread on LB agar plates with the appropriate antibiotics.

### 2.2. Determination and Separation of the Circular ssDNA in Vivo

For verify the activity of RepH protein and RCORI cassette, overnight culture of DH5α harboring plasmid pRC02 was inoculated to fresh LB medium with 100 µg/mL ampicillin in 1:100 dilution and harvested after 6 h induction at 37 °C with shaking. The detail of plasmids construction was shown in Appendix A. The purified plasmid DNA was detected on an agarose gel and the appropriate pRC03 band was cut and purified. The resulting purified pRC03 plasmids were then transformed into DH5α competent cells. A single colony was inoculated overnight into 5 mL of LB medium, containing 100 µg/mL ampicillin. The overnight culture was diluted 1:100 into fresh LB broth with appropriate resistance and grown until stationary phase. The plasmid was purified and assessed through the restriction analysis (ScaI).

For circular ssDNA verification analysis, PCR and enzyme digestion analysis was introduced to detect the existence of circular ssDNA. The pRC08 plasmids were purified and used as templates for PCR. Unless otherwise stated, 50 μL PCR reactions were performed using EasyTaq® DNA Polymerases (TransGen Biotech, Beijing, China) and the primer set PCR-2K-F/R (Table A1). Thermocycled reactions were initiated for 5 min at 94 °C and cycled 30 times, involving denaturation for 30 s at 94 °C, annealing for 30 s at 55 °C and primer extension for 30 s per kilobase at 72 °C, finally, the reaction was terminated at 72 °C for 5 min. The PCR product was recovered using plus DNA extraction kit and sequenced by GENEWIG. For enzyme digestion analysis, 10 µL (1 µg) of purified DNA was digested at 37 °C in a solution containing each BamHI and HindIII of 1 µL (= 5 U), 5 µL of restriction buffer (10× Flycut buffer) and 33 µL of ddH2O. Thereafter, the digested sample serves as a template for PCR and the PCR was performed as above.

For circular ssDNA determination analysis, fluorochromes probe binding assays was performed to determine circular ssDNA in the cell lysate. Three kinds of fluorescent probe were designed, which were specifically combined with the corresponding target sequence (Appendix A and Table A2). Overnight culture of DH10β cells harboring plasmid pRC17 were inoculated to fresh LB medium with 100 µg/mL ampicillin in 1:100 dilution and harvested at mid-log phase (OD600~1.0). After incubation, the circular ssDNAs were purified using cell lysis protocol (Appendix A). The hybridization assay took place in the same buffer as pre-experiments, the probe concentration was 10 nmol. The solutions were incubated at 30 °C for 1 h. The hybridization solution was then run on a 1.2% agarose gel and the appropriate band was judged by control.

### 2.3. Iterative Genome Editing Procedure

Genome editing efficiency with linear ssDNA was characterized that produced premature stop codon in the chromosomal lacZ gene. In general, 100 μL 10^−6^ diluted bacterial solution was plated on LB agar plates containing IPTG and 5-Bromo-4-chloro-3-indolyl β-D-galactopyranoside (X-gal) and grown overnight. Efficiency of allelic replacement was calculated by taking the ratio of the number of white colonies to the total number of colonies on plates. A similar strategy was used in insertion 1011 bp sequences experiments to determine the edit capacity of linear ssDNA. Correct editing colonies were amplified and sequenced by colony PCR with primer set PCR-MG1655-F/R for 11 bp substitution and primer set PCR-1011-F/R for 1011 bp cassette insertion. All experiments were conducted in triplicate and the data were presented as mean values ± standard deviation.

The genome editing efficiency was characterized by transforming the two plasmids into MG1655 cells. The linear donor DNA was provided by plasmid in vivo. The circle ssDNA with specific cleavage site was produced and cut by Cas9 protein. This approach could produce any linear ssDNA of designed length and sequence. To verify the genome editing was performed by linear ssDNA, three plasmids were constructed and characterized that produced four premature stop codons in the chromosomal *lacZ* gene. The pRC22, pRC20 or pRC11 was added to electrocompetent cells (harboring plasmid pRedCas9) for each 50 μL electroporation reaction, respectively. After plasmid transformation, 500 μL cells were inoculated into fresh LB media until OD reached 1.0. Then, the cells were diluted (1:10) into fresh LB media (1:100) for subculture with appropriate selective condition. Sequencing was performed to confirm that the four stop codons was introducing in *lacZ* gene. A similar strategy was used in the integration experiments that inserted 1011 bp sequence on *lacZ* gene to determine the genome editing capacity for linear DNA cassettes in vivo.

### 2.4. Quantitative and Statistical Analysis

All experiments were conducted in triplicate and the data were presented as mean values ± standard deviation.

## 3. Results

### 3.1. Engineering Rolling Cycle Replication for in Vivo Circular ssDNA Synthesis

The pRC02 plasmid was constructed to test the activity of the RepH protein and rolling circle origin (RCORI) in *E. coli*. The gene RCORI105 contains a 105 bp sequence which is necessary for the function of small plasmid replication origins, including the minimal replication origin of pC194 [35]. Efficient termination occurred at a 36 bp sequence (RCORI36) derived from pC194, which lacks a larger palindrome (14 bp) and was reported to have maximal termination activity (Figure 1A) [35]. RepH was placed under the control of prhaBAD so that it can be induced with L^+^Rha. The RCR starts when RepH binds to the RCORI105 sequence of the pRC02 plasmids and creates a 3’ end as the primer for replication. RepH remains covalently attached to the 5’-end of the nick site. Thus, by designing a RCORI36 sequence at the end of the RCR replicon, the small circular plasmid pRC03 was generated in vivo (Figure 1B). The small plasmid continues to replicate in *E. coli* by the θ mechanism in the presence of ColE1, and is selected via ampicillin resistance selection. Electrophoresis of plasmid DNA extracted from pRC02-DH5α cells showed that the yield of pRC03 was sufficient to observe a slight band of the correct length on the agar gel (Figure 1C). The recovered plasmid was used to transform DH5α competent cells, which were cultured in LB medium. Thereafter, the plasmid DNA was extracted and verified using *Sca*I restriction analysis. The band on the agarose gel was in agreement with the size of pRC03 (Figure 1D), and the recovered band was sequenced by sanger sequencing. The sequence analysis revealed that the isolated small plasmid pRC03 was identical with the prediction (Table A2). This result confirmed that circular ssDNA synthesis is functional in vivo.

Next, a series of plasmids were designed to demonstrate the concept and test the ability of our system for circular ssDNA production in vivo (Figure 2A). First, the plasmid pRC08 was constructed to produce a circular ssDNA of 1959 nts (approximately 2 kb; Appendix A). In addition to the RCORI and *repH* gene that were unchanged from pRC02, a 791 bp fragment of M13 was added into the plasmid. The fragment contained gene *V* and gene *X*, preceded by a strong promoter that was reported to increase circular ssDNA yield in vivo [34]. The plasmids were harvested and digested by *Bam*HI and *Hind*III. An assay was developed based on the PCR amplification using EasyTaq® DNA Polymerase. The plasmids and digested plasmid samples were used as templates to amplify the target products in the same PCR system. Comparative analysis of the PCR products clearly showed that circular ssDNA was generated in vivo (Appendix A). The size of the target band was 1872 bp, and treatment with restriction enzymes revealed a strong band. Sequencing of the band by sanger sequencing confirmed that the ssDNA was circular (Appendix A). Next, different plasmid vectors were tested to verify the yield of circular ssDNA. We introduced the plasmid pRC07, which was constructed based on pET19b, to express the RepH protein using the strong T7 promoter. After incubation under the same culture conditions, PCR was performed to compare the circular ssDNA yield (581 nts circular ssDNA) of pRC06 and pRC07 (Appendix A). The results showed that the pMVvector was a better choice for circular ssDNA production. After optimizing the vector, pRC09 was constructed to produce a circular ssDNA of 4974 nts (5 kb) in *E. coli* (Appendix A). The cells of *DH10β* and *DH5α* harboring pRC09 were disrupted using cell lysis solution (CLS), and the target sequence amplified to verify the circular ssDNA yield. This result showed that a longer ssDNA could be produced in vivo with a similar yield. The relative yield and ratio of circular ssDNA were evaluated by comparing the PCR products of the circular ssDNA, plasmid and the 16S rRNA standard. An assay was developed based on the capability of pRC06 and pRC08 to produce circular ssDNAs of different length. We quantified the concentrations of PCR products using the gray-scale intensity of scanned bands (Appendix A). The PCR yield of the circular ssDNA was only about 3–4 percent of the yields of the plasmid and 16S PCR products.

Based on these results and a pC194 model, we designed a new plasmid which included a 65 bp terminator (RCORI65) and *EcSSB* gene to increase the capacity of circular ssDNA production in *E. coli*. The RCORI65 sequence contains the replication origin of the ssDNA plasmid pC194 [37]. We developed an analytical protocol to amplify circular ssDNA products from cell lysates using PCR, which can be visualized using agarose gel electrophoresis. Two primer pairs were introduced to confirm the presence of RCORI105, and if the RepH protein preferentially identified it rather than RCORI65 as the replication origin. The experimental results were in agreement with the expectations. When RICORD105 was used as the RCR replication origin with RICORD65 as the terminator, the yield of circular ssDNA was higher than with RICORD36 as the terminator (Figure 2B), and its initiation activity was negligible. This was further confirmed by passaging experiments (Appendix A). To verify the sequence, the PCR band was separated by gel electrophoresis, purified and sequenced using a corresponding primer (Appendix A). A comparison of the quantities of PCR products from cell lysates obtained using different cell disruption methods showed that the newly designed plasmid had a better ssDNA production capacity. Although the cell disruption methods were different, the results were consistent (Appendix A). The advantage of using EcSSB as a single-strand binding protein for the circular ssDNA is that it is the endogenous ssDNA-binding protein of *E. coli.*

Taken together, the results indicated that the circular ssDNA yield could be increased by using pRC17. We next sought to determine the requirements for ssDNA synthesis using RCORI and RepH in *E. coli*, and constructed a series of plasmids with deletions of each element (Figure 2D). PCR was performed to verify the yield of circular ssDNA and confirm the sequence-specificity. The expression of RCORI105 or RepH protein alone was unable to produce the circular ssDNA because of the specificity of RepH protein for RCORI. By contrast, strong target band was observed in the presence of both the RCORI and RepH protein. This further confirmed that circular ssDNA can be produced only when all replication elements are present (Figure 2E). The expression of all three genes (RCORI, RepH protein and EcSSB protein) enhanced the production of the circular ssDNA eightfold compared to the combination of RCORI and RepH protein. The plasmid pRC17 was then used to produce circular ssDNA in *E. coli* DH10β, which lacks the intracellular exonuclease and SOS response. This plasmid included RCORI105, RCORI65, RepH protein and a strep-tagged EcSSB protein. The simple cell disruption methods of freeze-thawing and enzymatic lysis by lysozyme were used to prepare the cell extract. To better characterize the circular ssDNA, we used a fluorescent probe binding assay to demonstrate that the circular ssDNA was indeed present in the cell lysate (Figure 3A). The binding of the fluorescence probes to DNA in the cell lysate was analyzed using the electrophoretic mobility shift assay (EMSA) in agarose gels. We firstly used the 90 nt oligo to test the specific binding between the probe and the target sequence (Appendix A). The oligo contained sequences complementary to probes 1 and 2, but identical with probe 3. After the binding assays, we validated the specific binding of the probe to the targeted DNA sequence in the presence of probes 1 and 2. It was clear that probe 3 could not bind to the 90 nt oligo, because the oligo has no complementary sequence with probe 3. Under different binding conditions, the binding efficiency was similar. Thus, the sample was incubated at 37 °C for 90 min.

Thereafter, a control group was introduced to confirm the size of the hybrid product containing the probe and the circular ssDNA. The fluorescence hybridization was intense and highly specific to the target gene. Three of the probes are visible in Figure 3B. The DNA in the cell lysate hybridized with probe 1 and showed a clear band (1984 nt circular ssDNA-probe) on the 1.2% agarose gel. Hybridization probe 2, which was hybridized with the plasmid and had no fluorescence band, confirming that the fluorescence band is a circular ssDNA product. These results visualized and further confirmed the nature of the circular ssDNA product via fluorescence probe binding.

The purification protocol was based on the specific binding between the strep-tagged EcSSB and Magrose Strep-Tactin beads, which were used to isolate circular ssDNA products from lysed cells (Figure 3C). The concentration of the recovered samples was sufficient to observe a slight band of the correct size (Figure 3D). To confirm that the band contained circular ssDNA, we recovered it using a gel extraction kit and amplified it by PCR (Figure 3E). The results clearly showed there was only circular ssDNA in the eluate. To verify the sequence, the 1897 bp band was gel purified and sequenced by sanger sequencing.

### 3.2. Homologous Recombination Using in Vivo Synthesized ssDNA Donor

Since the initial data strongly suggested that circular ssDNA could indeed be produced in vivo, we converted the circular ssDNA to linear ssDNA in vivo and used it directly for genome editing (Figure 4A). For this objective, Spy Cas9 protein and a corresponding guide RNA (gRNA) were introduced. It has been demonstrated that the Cas9 protein can recognize and cleave single-stranded DNA (ssDNA) by an RNA-guided, PAM-independent recognition mechanism [38]. Furthermore, the Spy Cas9 enzyme is highly selective for its cognate guide RNA and only supports dsDNA or ssDNA cleavage when its own guide RNA is used in the reaction [39]. The results suggested that Cas9 protein nicked the circular ssDNA under the guide RNA without a PAM site (Appendix A), and converted it into linear ssDNA. The guide RNA was carefully designed to avoid recognizing homologous sequences within the genome of *E. coli* MG1655. To appropriately design the sequences of our guide RNA (Appendix A), we used the nucleic acid structure prediction tool Nucleic Acid Package (NUPACK) [40].

Firstly, we used 90 nt oligos to test the efficiency of genome editing, whereby one donor was not completely complementary to the target sequence at the 3’ and 5’ ends (Figure 4B). The two 90 nt oligos were introduced to produce mismatch changes in a targeted region of the *lacZ* gene in two distinct cell populations. Blue-white colony counting was used for preliminary statistical analysis of the editing efficiency. Through successive cycles of MAGE, the chromosomal sequence of the *lacZ* region increasingly diverged away from the wild type. The editing efficiency was quantified after MAGE cycle 4, which showed no significant difference (*p* = 0.23) in editing efficiency (around 12%) between the two groups (Figure 4B). Then, the targeted genomic region of the white colonies was amplified by PCR and sequenced to confirm the validity of genome editing.

Genome editing using the linear ssDNA system is determined by a combination of three factors: the guide RNA for Cas9 protein, the RCR system for the production of the donor circular ssDNA, and the λ-Red homologous recombination system. Under the tested experimental conditions, the genome editing was effective. Three different plasmids (Figure 4C) were separately introduced into MG1655 cells harboring pRedCas9. In passage-1, pRC22 produced a white colony on the plate, which indicates the generation of a correctly edited colony within 12 h (Figure 4D). Through successive passaging, the chromosomal sequence of the *lacZ* region increasingly diverged away from the wild type (Figure 4D, Appendix A). The rate at which pRC22 induced genomic changes progressively increased from 0.73% (passage 1) to 41.45% (passage 10) (Figure 4E). By contrast, pRC11 showed 0% editing efficiency even at the 10th passage. The pRC20 plasmid led to a gradual increase of correct editing from 0% to 0.73%. Because our screening is not selective, the calculation of editing efficiency here is based on the total number of bacterial cells. The sequencing results from amplified products in passage-1 and passage-10 indicated that the correct editing rate was 100% in passage 1, but passage-10 had only around 60–80% correctly edited colonies. However, the incorrect colonies were accounted for by 2–4 false positives or colonies that yielded no PCR products. Given the scalable nature of our approach, further increase of the insert at the identical genome editing site to 1011 bp did not significantly affect the number of correctly edited colonies (Figure 4F). We detected insertions by amplifying the target region followed by Sanger sequencing. These results collectively confirmed that the RC-Cas mediated genome editing system allows the efficient integration of DNA as large as 1 kb.

## 4. Discussion and Conclusions

In this study, we developed a scalable platform for genome editing based on in vivo synthesis of linear ssDNA in living cells. The system utilizes a modular rolling circle replication structure, which is converted into linear ssDNA via cleavage by the PAM-independent SpyCas9 nuclease. By comparing the ratios of blue/white colonies obtained after targeting the chromosomal *lacZ* gene with three different plasmids, we confirmed that the RC-Cas-mediated linear ssDNA integration system could effectively improve the efficiency of genome editing. Notably, the editing efficiency was calculated based on the entire population of bacteria, rather than using a survival-based selection step. Correct editing was only observed in the experimental group in passage 1, and the efficiency was 0.73%. For verification of the editing capability, a 1011 bp sequence was inserted into the genome and sequencing was used to confirm that the sequence was inserted in the correct location. It is noteworthy that the ssDNAs in our system have been used directly for genome editing without additional preparation steps.

The concentration of donor DNA is another crucial issue influencing the efficiency of genome editing, but this could be addressed by improving the efficiency of the RCR module and developing a more suitable method for the linearization of circular ssDNA in vivo. Previous studies revealed that many factors in *E. coli* play important roles during pC194 replication. For example, the production of ssDNA can be effectively increased (up to 70% of the total plasmid DNA) by the overexpression of a single-stranded DNA-binding protein (M13 gene *V*) in the cell [34]. Similarly, the deletion of rriB from plasmids can effectively prevent the transformation of circular ssDNA into the dsDNA form [34]. UvrD can act as a helicase during the replication of pC194 in *E. coli* [41]. Consequently, we examined several parameters to improve the efficiency for circular ssDNA production in *E. coli*. The highest ssDNA yield was achieved using stable modules (RCORI105 and RCORI65) combined with the expression of RepH and EcSSB proteins. Implementing this requires that the terminator ensures that the circular ssDNA ends precisely at RCORI65 after replication. We also confirmed that the production of circular ssDNA could be increased by overexpressing an ssDNA binding protein. The EcSSB protein is the endogenous ssDNA-binding protein of *E. coli* and is known to protect ssDNA during plasmid replication [42]. All ssDNA production experiments were performed in the *E. coli* strain DH10 beta, which lacks intracellular exonuclease activity and SOS response, thus preventing the ssDNA degradation [43]. To our best knowledge, this is the first study demonstrating that the rolling cycle replication of pC194 is functional in a commonly used *E. coli* strain, while other studies relied on specifically constructed laboratory strains [34,35]. However, the efficiency of circular ssDNA synthesis was strain-dependent, whereby DH10 beta was better compatible than MG6155.

In addition, we consider that the conversion efficiency of circular ssDNA to linear ssDNA is also important for genome editing in vivo. This might be solved by introduced other mechanisms for DNA cleavage, such as DNAzymes [44,45] or CRISPR-Cas12a proteins [46,47]. Here, we used the RNA-guided, PAM-independent recognition mechanism of SpyCas9 to recognize and cleave circular ssDNA [38]. Due to this mechanism, the Cas9 nuclease complex only cleave the circular ssDNA in complex with the gRNA, while it does not damage the dsDNA plasmid vector. This is very convenient for engineering, since SpyCas9 can be combined with different gRNAs to specifically cleave genomic dsDNA as well as the in vivo synthesized ssDNA. The genetic modifications were successfully introduced in the cell population probably because of the RC-Cas-mediated platform directing synthesized ssDNA to the lagging strand of the replication fork during DNA replication. Further exploration of this platform is required to determine the exact mechanisms involved in ssDNA synthesis and integration into genomic DNA. Another potential benefit of our platform is that the desired ssDNA can be integrated into the cells and used directly in gene editing without additional expenses. The cost of commercially synthesized oligos currently ranges from $0.10 to 0.30 per bp ($100–300 for a 1kb gene) [48]. Consequently, approximately $30 of synthesized oligos are needed to introduce 27 bp of mutations at full degeneracy for a single genomic target [13]. Particularly, it demonstrated that circular ssDNA may be a suitable intermediate material in living cell for bioengineering, due to liner ssDNA generally being considered with important role as signal for cell damage and rising many molecular rescue reactions [49,50]. In view of replication intermediate ssDNA of many bacteria plasmids or phage was converted to dsDNA from specific loci [23,37], it may implicit that in nature the circular ssDNA may couple with more in vivo DNA metabolism pathways and genomic element transfer. Furthermore, because there were only two proteins, RepH and Cas9, necessary for this in vivo ssDNA generation, we believe that it is very possible to be applied on mammal cells; however, there will be more demand in bacteria cell engineering with large genome modification for bio-product synthesis in highly human designed way. To the best of our knowledge, this is the largest ssDNA structure synthesized in vivo for genome editing to date. We hope that this RC-Cas-mediated on-site ssDNA synthesis system will become a useful molecular tool that is compatible with most genome editing system as donor template, and pave the way for dynamic genome editing applications.

## 5. Patents

The initial filings were assigned Chinese patent application 201911382846.5.

## Figures and Tables

**Figure 1 cells-09-00467-f001:**
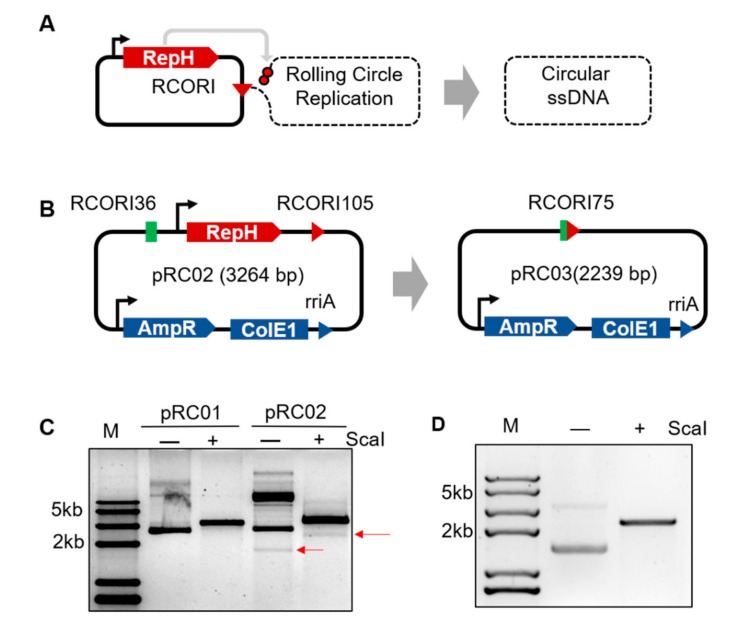
The mechanism for rolling circle origin (RCORI)-mediated DNA synthesis in *Escherichia coli*. (**A**) Schematic of circular single strand DNA (ssDNA) production in vivo. The RCR initiator protein RepH, which acts as a dimer, binds to the RCORI through a sequence-specific interaction. This is followed by nicking of the RCORI by initiator protein (RepH), unwinding of the DNA by helicase, and binding of the single-strand DNA-binding protein (SSB) to the displaced leading strand. Once the replication fork reaches the termination site, the free monomer of RepH cleaves the displaced ssDNA. A series of regulation and cleavage events follow, resulting in the release of a circular ssDNA (displaced leading strand). Solid lines represent double-stranded DNA and dashed lines represent single-stranded DNA. (**B**) Construction of the small plasmid. RCORI105, the RCR origin, sequence consist of 105 bp from pC194; RCORI36, the 36 bp RCR termination sequence from pC194; AmpR, ampicillin resistance cassette; ColEI, high-copy-number ColE1 origin of replication; RCORI75, the newly formed RCORI after circular ssDNA release. (**C**) Generation of the small plasmid pRC03. pRC01 is a control group which not including RCORI36 to verify small plasmid construction. The arrows show the band of the small plasmid. *Sca*I was applied to verify plasmid length. Lane M, DNA Marker. (**D**) Validation of the small plasmid after recovery. -, without ScaI; +, with ScaI. The full sequence of the small plasmid is shown in Table A2.

**Figure 2 cells-09-00467-f002:**
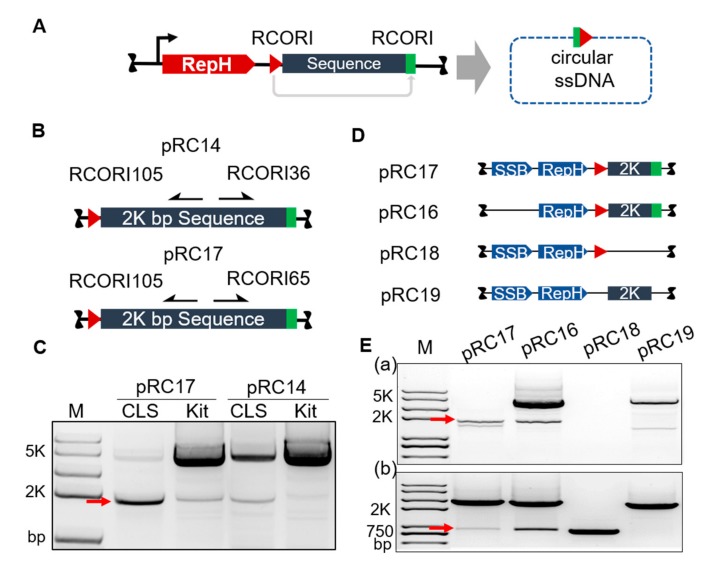
Design of genetic parts for circular ssDNA production in vivo (**A**) Schematic for the conversion of the customized gene to ssDNA. The main RCR sequence contains the *repH* gene, the desired ssDNA sequence and the RCORI gene. Solid lines represent double-stranded DNA and dashed lines represent single-stranded DNA. (**B**) The structure of pRC14 and pRC17 used to produce circular ssDNA in vivo. The only difference between these two plasmids is the RCR terminator, which is RCORI36 in pRC14 and RCORI65 in pRC17. The arrows represent the designed amplification primers of the circular ssDNA. (**C**) The terminator strengths of RCORI36 and RCORI65. The circular ssDNA yield was measured by comparing the abundance of the corresponding PCR products. CLS, cells lysed using a lytic buffer; Kit, plasmids extracted using a commercial miniprep kit; the arrow shows the PCR product (1978 bp) of the circular ssDNA. (**D**) Different combinations of the RCR system. Lane M, DNA Marker (**E**) The impact of different combinations on the circular ssDNA yield and the effect of RCORI65 as an RCR origin. (**a**) The expected PCR product for testing the necessary components to produce ssDNA is 1978 bp. (**b**) The expected PCR product for testing RCORI65 as an RCR origin is 778 bp. The plasmid pRC17 is a control containing all of the RCR components.

**Figure 3 cells-09-00467-f003:**
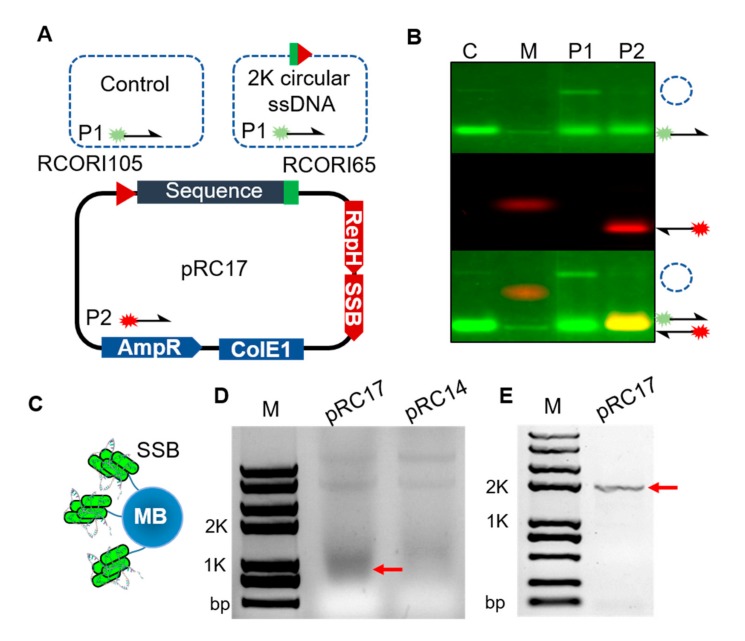
Verification and separation of circular ssDNA. (**A**) The mechanism of the probe binding with the target sequence. Probe 1 is complementary to the circular ssDNA and the control group; probe 2 is complementary to the plasmid. (**B**) Fluorescence images of the probe binding with DNA from the cell lysate. Lane C, probe 1 + control; Lane P1, probe 1 + cell lysate; Lane P2, probe 2 + cell lysate; Lane M, DNA Marker (**C**) Schematic of the ssDNA recovery using Magrose Strep-Tactin beads. (**D**) Comparison of the circular ssDNA recovery yield from cell lysate. Data are shown for the production of circular ssDNA under culture conditions for plasmid extraction. The arrow indicates the band of the circular ssDNA. (**E**) PCR analysis of the recovered circular ssDNA. The arrow shows the PCR products (1978 bp) of the circular ssDNA.

**Figure 4 cells-09-00467-f004:**
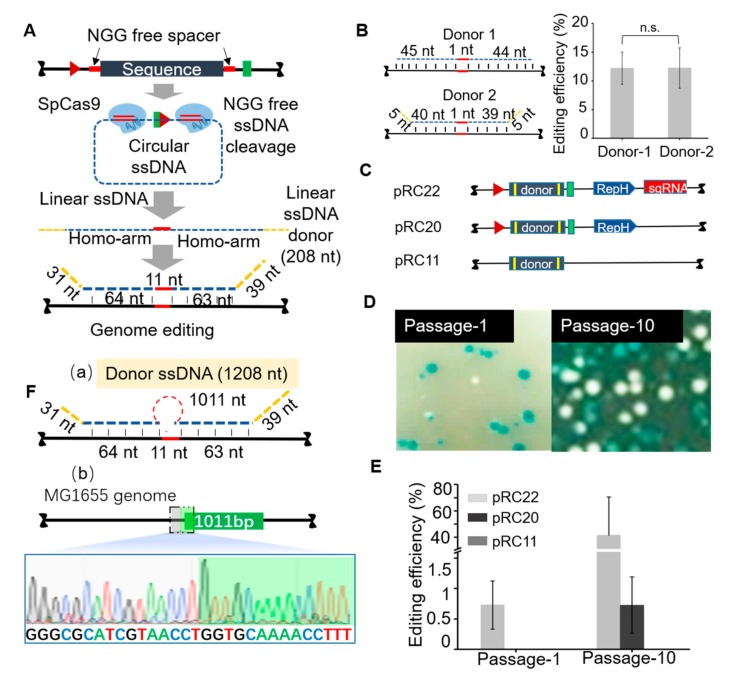
RC-Cas-mediated genome editing in vivo. (**A**) The mechanism of the production of linear ssDNA. SpCas9-mediated DNA cleavage of the targeted circular ssDNA, in the absence of a protospacer adjacent motif (PAM) sequence. (**B**) Verification of the allelic replacement efficiency. There was no significant difference (*p* = 0.23) in the replacement efficiency between donor 1 and 2 (around 12%) (**C**) The plasmids with different components used for gene editing. pRC22 includes the RCR system, gRNA and donor DNA; pRC20 includes the RCR system and donor DNA; pRC11 only includes the donor DNA (**D**) Photograph of the plates showing the effect of linear ssDNA synthesis for gene editing using pRC22. Passage 1, the first round of culture for introducing the allele substitution (11 bp substitution in the *lacZ* gene); Passage 10, the tenth round of culture for introducing the allele substitution (11bp substitution in the *LacZ* gene). (**E**) The efficiency of substituting 11 bp in the *LacZ* gene using the linear ssDNA donor (**F**) Verification of the insertion efficiency of 1011 bp. (**a**) The mechanism of the 1011 bp insertion (**b**) Sanger sequencing of the 1011 bp insertion.

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
