# Peer review of "Dynamic Genome Editing Using In Vivo Synthesized Donor ssDNA in *Escherichia coli"

_cells, 2020, doi:10.3390/cells9020467_

Round 1
Reviewer 1 Report
Review
Cells-701963
“Dynamic genome editing using in vivo synthesized donor ssDNA in Escherichia coli”
Hao et al.
Dear Authors
The following article provide a comprehensive technical guide, demonstrating an improved method for generation of donor ssDNA sequences “in vivo” that could be processed by using CRISPR-Cas9 nuclease and combined with lambda Red recombinase could be implicated as donor for precise genome engineering. The presented system utilizes a modular rolling circle replication structure, which is converted into linear ssDNA via cleavage by the PAM-independent SpyCas9 nuclease. From the other hand, by comparing the ratios of blue/white colonies obtained after targeting the chromosomal lacZ gene with versatile set of plasmids, the authors have demonstrated that RC-Cas-mediated linear ssDNA integration system could competently improve the efficiency of genome editing. Moreover, all of the presented experiments were performed in E. coli strain DH10 beta, which to the best of authors knowledge, renders the current study, the first one demonstrating functional rolling cycle replication of pC194 in a commonly used E. coli strain.
The technical details provided by the authors are very well presented and could serve as a comprehensive guide in future gene editing experiments. There are also, no major weak points in the study, which may jeopardize the drawn conclusion.
However, despite all superlatives I would like to express few general concerns, which might help the paper to deliver better its content and to reach the proper audience.
Despite all technical improvements, which the paper potentially could deliver, there is still a question about the practical relevance of the current study. It is true that the CRISPR-Cas9 technology opens unlimited possibilities for gene editing and everything along those lines should be considered of a great interest, but in the following study there are no evidences that the demonstrated RC-Cas-mediated linear ssDNA integration system could evolve and could be developed to a level compatible with its implication in eukaryotic model organisms, where CRISPR/Cas9 technology really “shine”. It will be great if authors expand on this in the revised version of the manuscript with any possible instrumentation. Moreover, the clearly technical aspect of the study and a very narrow and detailed molecular cloning profile of the paper, render the presented article out of the scope of the Cells journal. My impression is that the article will fit much better in journals like, Bioengineering or Microorganisms, which are also part of the MDPI publishing group, as their aims and scopes much accurately match the problems, current article is trying to address. Authors have pointed out that the further exploration of the described platform is required to determine the exact mechanisms involved in ssDNA synthesis and integration into genomic DNA. Indeed, I agree with this statement and suggest authors to share at least some speculation concerning the above statement.Author Response
Please see the attachment.Thank you.

Reviewer 2 Report
This paper describes a novel method for generation of long and circular single-stranded DNA (ssDNA) in bacteria (E. coli), which will be used for targeted knock-in (KI) experiments. ssDNA has been used as donor DNA (template) for homology direct repair (HDR)-mediated KI. In almost cases, short ssDNA that is chemically synthesized in vitro has been used, but for more precise KI employment of large-sized ssDNA (> 1 kb) is desirable. At present, a few methods for its synthesis are available. The authors aimed to provide an alternative to acquisition of such longer ssDNA in vivo by using bacteria. In this context, this approach appears to be valuable and interesting. However, this paper has numerous errors in description and is associated with the difficulty to understand/grasp the entire procedure of its technology, probably in part due to the lack of explanation of this system in an understandable manner. For example, in Abstract section there is no explanation of “RC-Cas-mediated” which requires formal terminology prior to use the abbreviation. This is also true for the case of ssDNA. In Fig. 1, labeling of “C” is mistaken, which should be labeled “D”. pRC01 is used as control, but there is no information about this plasmid. The authors mentioned “Expression of the small plasmid” in the head of (C), but this expression is inappropriate. “Generation of the small plasmid pRC03” is preferable for the readers.
What we wish to know is the total amount of circular ssDNA produced from the transformed E. coli. Unfortunately, there is no description about this. Furthermore, the authors checked the fidelity of this resulting ssDNA using the targeted KI experiment in E. coli. However, the main concern of many scientists would be whether this in vivo-synthesized ssDNA is applicable to the KI experiments using cultured mammalian cells or early embryos. In this context, it is unfortunate that these issues have not been assessed in this paper.
Reviewer 3 Report
Hao et al. describe a novel system to produce large sequences of ssDNA in E. Coli in vivo. In combination with SpyCas9 this ssDNA can be used for genomic engineering mediated by lambda Red recombinase. This is an interesting technique for introducing specific mutations into bacteria. There might be additional future applications for large ssDNA sequences in bacteria, where this novel technique could be of help.
Comment:
It would be interesting if the authors could give practical examples for biological questions that could be addressed with their technique. This helps in understanding its value and might also promote its use.
Round 2
Reviewer 2 Report
When I saw the revised version of paper, some are indeed corrected, but almost part of this paper still has numerous errors.
My opinion to this paper is that it would be better to provide a Supplementary Figure, in which RC-Cas-mediated amplification of double stranded DNA in E. coli is described in more detail and comprehensive manner. Also, it may be required for explanation on a series of pRC vectors.
As another question, I would like to know the total amount of circular ssDNA produced from the transformed E. coli. This point appears not to be mentioned in the revised manuscript.
Furthermore, many researchers involving in genome editing work appear to obtain more rigid and promising approach for HDR-mediated KI of a gene of interest. In this sense, long and circular single-stranded DNA (ssDNA) synthesized in E. coli should be valuable in that field. Unfortunately, no data on successful application of this system to mammalian cells or early murine embryos have been provided yet.